# Celiac Disease Genetics, Pathogenesis, and Standard Therapy for Japanese Patients

**DOI:** 10.3390/ijms24032075

**Published:** 2023-01-20

**Authors:** Tasuku Tamai, Kenji Ihara

**Affiliations:** Department of Pediatrics, Faculty of Medicine, Oita University, 1-1 Idaigaoka, Yufu 879-5593, Oita, Japan

**Keywords:** celiac disease, Asia, Japan, genetics, pathogenesis, therapy

## Abstract

Celiac disease is an autoimmune disease primarily affecting the small intestine that is caused by the ingestion of gluten in genetically susceptible individuals. The development of celiac disease is based on a complex immune response to gluten proteins. The global average prevalence in the general population is about 1%. In recent years, it has become clear that celiac disease is not less common in Asian countries than in Western countries but often remains undiagnosed. Although the number of patients with celiac disease in Asia is expected to increase with improving disease recognition and advances in diagnostic techniques, there remain few reports of celiac disease in the Far East region of Asia, especially in Japan. In this paper, we outline the epidemiology, diagnosis, and treatment of celiac disease. In addition, we summarize the reported Japanese cases of celiac disease with an overview in Japan.

## 1. Introduction

Celiac disease (CD) is a gluten-related disorder (GRD), a term that refers to a range of conditions caused by the ingestion of gluten proteins in wheat, barley, and rye. GRDs include not only CD but also dermatitis herpetiformis and gluten ataxia, which are considered autoimmune diseases. Wheat allergy is considered an allergic disease, whereas nonceliac gluten sensitivity is considered a nonallergic and nonautoimmune disease [1,2] (Figure 1). CD is an immune-mediated inflammatory disease of the small intestine caused by gluten proteins ingestion in individuals with a genetic predisposition to human leukocyte antigen (HLA) DQ2 or DQ8. Ingestion of these proteins promotes immune-mediated mucosal inflammation of the proximal small intestine with villous atrophy and crypt hyperplasia, leading to malabsorption and gastrointestinal symptoms [3,4,5,6]. A gluten-free diet is the standard treatment, and removal of gluten from the diet improves intestinal lesions and symptoms. The prevalence of this disease is reportedly high in Europe but low in Asia. Among Asian regions, the countries with the lowest incidence of CD are mainly located in the Far East; indeed, CD is as common in China as in the West but relatively rare in Korea and Japan [7,8,9,10]. To our knowledge, only 27 Japanese cases of CD and seropositivity have been reported [11,12,13,14,15,16,17,18,19,20,21,22,23,24] (Table 1).

## 2. Epidemiology

### 2.1. General Population

CD originally exclusively affected Caucasian Europeans but is now globally distributed [3,25]. In many countries, the overall prevalence of CD in the general population ranges from 0.5% to 2%, with an average of approximately 1% [3,25]. CD can occur at any age, from childhood after two years old to the second and third decades or later of life, showing a female predominance [4,26]. The global distribution of CD is thought to parallel the distribution of HLA genotypes susceptible to CD, provided that the population is also exposed to gluten [27]. As mentioned above, prevalence is higher in women than in men, with women being affected approximately 1.5 to 2 times more often than men [7,28]. The prevalence of CD is also higher in children than in adults (0.9% vs. 0.5%) [7]. According to one systematic review and meta-analysis, the pooled prevalence of CD from 1991 to 2000 was 0.6%, and that between 2011 and 2016 was 0.8%. These results suggest an increase in the prevalence of CD over time [7]. Since the late 20th century, the introduction of noninvasive and accurate CD serologic tests has made the diagnosis of CD more efficient [29]. For example, the incidence of pediatric CD in Canada increased threefold after the introduction of endomysial antibody (EMA) tests [30]. However, while CD diagnostic techniques and practice guidelines have been updated over time and awareness and knowledge of CD has been increasing, up to 95% of CD patients reportedly remain undiagnosed [31,32]. Some studies have found that the delay in the diagnosis of CD ranges from 4 to 10 years [33,34]. Even in developed countries, there are many undiagnosed cases, with most showing atypical signs or vague symptoms [35]. Limited access to diagnostic serological tests and a lack of experienced specialists may be the reason for the delayed or missing diagnoses in developing countries [36]. 

### 2.2. CD in Asia and Japan

A systematic review and meta-analysis of population-based studies that included 275,818 subjects showed that the pooled global seroprevalence of CD in the general population was 1.4%, and the biopsy prevalence of CD in the general population was 0.7% [7] (Table 2). CD had been considered to be uncommon in Asia for a long time, but several studies published in the past two decades have demonstrated that CD is present and as prevalent in the Indian subcontinent and Middle East as in Western countries [37,38,39]. The seroprevalence of CD in Asia is 1.8%, and the biopsy prevalence is 0.6% [7] (Table 2). The seroprevalence and biopsy prevalence of CD in Asia reported by Ashtari et al. was originated from 11 Asian countries, as follows: Turkey, India, Iran, Israel, Saudi Arabia, Arab Emirates, Kuwait, Oman, Malaysia, China, and Japan. Based on these findings, we divided these countries into three geographical regions: Middle East (Iran, Israel, Saudi Arabia, Arab Emirates, Kuwait, Oman, and Turkey), South Asia (India and Malaysia), and East Asia (China and Japan). The three categorized countries (seroprevalence among general population) were Middle East (1.4%), South Asia (1.2%), and East Asia (0.06%) (Table 2). The low seroprevalence in the East Asian general population was attributed solely to the Chinese report. The biopsy prevalence among the general population was 0.59% in the Middle East, 0.87% in South Asia, and 0.05% in East Asia (Table 2). The biopsy prevalence in the general population of East Asia is low based on the only available report from Japan. Asian countries with a higher biopsy prevalence included India (0.3–1.4%), Israel (0.6–0.7%), Turkey (0.3–0.5%), and the Central and West Asian regions [40,41]. Among the Far East countries of China, Korea, and Japan, more case reports of CD have been published from China than from other countries. Although East Asia has a lower prevalence of CD than other Asian countries, it has also been reported that China has a prevalence similar to that of Europe and the United States [10]. In China, seroprevalence and biopsy prevalence were 1.27% and 0.35% among ethnic groups in the Xinjiang Uighur Autonomous Region [42]. Comparing the prevalence of CD among Chinese living in urban and rural areas, there is a large difference. The prevalence was three times higher in rural areas with high wheat consumption than in urban areas [42]. Unlike China, there are few reports from South Korea and Japan. Indeed, in South Korea, only three CD cases have been reported [43,44,45]. Choi et al. reported that 1 out of 76 Koreans who had undergone antitissue transglutaminase (tTG) antibody tests in the past 9 years had positive test results, and even then, the tTG antibodies in that patient were only faintly positive (less than twice the upper limit of normal). This case should have been confirmed to be CD using an intestinal biopsy, but information on further diagnostic approaches for CD, including any biopsies performed, was lacking, so the definite diagnosis of CD remains unclear [9]. In Japan, the epidemiological frequency of CD is extremely low [8]. Fukunaga et al. described only two biopsy-based CD cases among 2055 people, including 2008 asymptomatic individuals and 47 adults complaining of chronic abdominal symptoms, which corresponds to a <0.1% prevalence [17]. The same research group reported that the seropositive rate (based on tTg antibodies ≥10 U/mL) was 0.19% in 2055 Japanese adults tested in 2008–2013 [15]. There were only 27 Japanese cases of CD and seropositivity (Table 1). Among Far East countries, reports of CD are particularly rare in both Korea and Japan. However, due to the worldwide recognition of CD and advancements in testing methods, the number of reports has increased globally, including in Japan and South Korea, since 2010 [11,12,13,14,15,16,17,18,19,20,21,22,23,24,43,44,45].

### 2.3. High-Risk Groups

The prevalence of CD is significantly increased in high-risk groups, including patients whose relatives have CD and those with Down syndrome, type 1 diabetes mellitus (T1DM), selective IgA deficiency, autoimmune thyroiditis, Turner syndrome, Williams syndrome, juvenile chronic arthritis, and inflammatory bowel disease (IBD). Only two cases of CD have been reported in Japan in patients with ulcerative colitis (Table 1). Because CD itself is rare in Japan, general practitioners often do not consider CD as a potential cause of these symptoms in at-risk patients; therefore, it is important for Japanese clinicians to consider CD as a differential disease and to perform CD-specific testing when chronic gastrointestinal symptoms are present in patients from high-risk groups.

#### 2.3.1. Patients Whose Relatives Have CD

Systematic reviews and meta-analyses have reported that the prevalence of CD is approximately 7.5% in patients who are first-degree relatives of others with CD and approximately 2.3% in second-degree relatives [46]. Female first-degree relatives showed a greater prevalence than male first-degree relatives (8.4% vs. 5.2%). The prevalence of CD was highest in siblings (8.9%), followed by offspring (7.9%) and parents (3.0%) [46]. A retrospective cohort study of 609 relatives from 1994 to 2016 was performed on 427 relatives (70%), noting a high prevalence of 15% [47]. Because of the high prevalence among first-degree relatives with CD, several guidelines recommend first-degree family screening for asymptomatic CD. HLA typing may be offered as a first-line test [6,48]. However, approximately 30% of symptomatic first-degree relatives of people with CD did not undergo the testing recommended by the guidelines [49].

#### 2.3.2. Down Syndrome

CD occurs in 1–19% of individuals with Down syndrome [50]. These individuals have a more than 18-fold higher incidence rate of CD than the general population [51]. A recent meta-analysis involving 4383 Down syndrome patients reported a CD prevalence of 5.8% [52]. The prevalence of CD was higher in studies that included only children with Down syndrome than in those that included both children and adults with Down syndrome. The prevalence of Down syndrome is independent of geographic location. The CD Guideline Committee of the North American Society for Pediatric Gastroenterology, Hepatology, and Nutrition (NASPGHAN) recommends CD screening for asymptomatic Down syndrome children [53], and the American Academy of Pediatrics recommends testing for CD in Down syndrome children with CD-related symptoms [54]. According to the latest recommendations of the European Society for Pediatric Gastroenterology, Hepatology and Nutrition (ESPGHAN) from 2012, HLA-DQ2 and HLA-DQ8 typing should be the first line of screening for CD in cases of Down syndrome [55].

#### 2.3.3. T1DM

The common genetic background of T1DM and CD is primarily based on the presence of HLA class II genes as DQ2 and DQ8. They are present in 95% of patients with T1DM and almost 99% of patients with CD (compared to 40% of the unaffected population), representing a significant risk factor for both diseases [4]. CD has a worldwide frequency of 1%, which rises to approximately 5% in T1DM patients, making CD an extremely frequent autoimmune disorder occurring in T1DM patients [56]. The prevalence of CD varies between 1% and 10% in patients with T1DM. In a cohort of children and adolescents with T1DM, the seroprevalence of CD was 15.4%, while the biopsy prevalence of CD was 6.9% [57].

#### 2.3.4. Selective IgA Deficiency

A total of 7.7% of individuals with selective IgA deficiency have CD [58]. Conversely, 2% to 2.5% of individuals with CD have IgA deficiency [55,59,60]. As such, because most specific serologic tests for CD involve IgA antibodies, patients with a known IgA deficiency require the application of special strategies to screen for CD [5].

#### 2.3.5. Autoimmune Thyroiditis

Up to 9.9% of individuals with autoimmune thyroiditis develop CD [61]. A pooled analysis based on 6024 autoimmune thyroiditis patients found a CD prevalence of 1.6%. The prevalence was higher in children with autoimmune thyroiditis (6.2%) than in adults (2.7%) and in studies examining both adults and children (1.0%). CD was also more prevalent in cases of hyperthyroidism (2.6%) than in those of hypothyroidism (1.4%) [61].

#### 2.3.6. Turner Syndrome

The prevalence of CD was shown to be high in cases of Turner syndrome. A total of 6291 patients with Turner syndrome were examined across 40 studies investigating the prevalence of CD, showing a crude CD prevalence of 3.8% [62]. The highest CD prevalence of 20% was reported in a study of 97 patients with Turner syndrome registered in the National Patient Register in Sweden [62].

#### 2.3.7. Williams Syndrome

Patients with Williams syndrome are known to have a high prevalence of CD [63,64,65]. Indeed, the prevalence of CD in Williams syndrome patients was 10.8% [64]. Although available data are limited, some experts recommend periodic testing in all cases of suspected CD among patients with Williams syndrome [65].

#### 2.3.8. Juvenile Chronic Arthritis 

The prevalence of CD was 2.4% in a retrospective cohort of 321 juvenile chronic arthritis patients who underwent CD screening in Italy from 2001 to 2019 (approximately 3 times the rate in the general population) [66]. There is no definite recommendation for CD screening in JIA patients according to the recently updated guidelines for the CD diagnosis and management [5]. 

#### 2.3.9. IBD

IBDs, including ulcerative colitis and Crohn’s disease, are associated with an increased risk of developing CD. One report analyzed 34,375 patients with IBD and reported that 0.93% of patients with IBD had CD, compared with 0.31% of non-IBD patients. The prevalence of CD was higher in childhood-onset IBD than in adult-onset IBD (1.66% for childhood-onset cases and 0.79% for adult-onset cases) [67]. A positive relationship between IBD and CD was found, similar to other reports [68]. 

#### 2.3.10. Cryptogenic Hypertransaminasemia

Patients exhibiting “cryptogenic hypertransaminasemia” should be screened for CD, as this condition may be the sole manifestation of silent CD [69]. The prevalence of CD in patients with unexplained hypertransaminasemia is 10%, a finding that justifies screening for CD in all patients with abnormal liver biochemical test results [70,71].

#### 2.3.11. Atopic Dermatitis

Atopic subjects are also another high-risk subgroup worth screening for a CD diagnosis [72]. CD sometimes runs a subclinical/silent course and is often associated with immunologic diseases. Atopic dermatitis is described as one of the most frequently associated conditions. A meta-analysis compared the prevalence of CD in AD and controls, showing a higher prevalence of CD [73].

## 3. Pathogenesis

Gluten is the most important environmental factor in CD. After its ingestion, gluten peptides cross the epithelial barrier and are deamidated by the enzyme tissue transglutaminase 2. Deaminated gluten peptides efficiently bind to HLA-DQ2/8 molecules on antigen-presenting cells, leading to the activation of gluten-specific CD4+ T cells. Gluten-specific CD4+ T cells secrete proinflammatory mediators, such as interferon γ, interleukin-2, interleukin-21, and tumor necrosis factor. As a result, the immune response causes inflammation and gut tissue damage [74,75,76]. Gluten-specific CD4+ T cells respond to gluten ingestion by releasing IL-2 and other inflammatory mediators, causing gastrointestinal symptoms, such as nausea and vomiting [77,78].

### 3.1. Genetic Factors

#### 3.1.1. HLA-DQ2 and DQ8

The genetic basis for CD has been shown by its frequent occurrence within families and its close association with HLA DQ2 and/or DQ8 loci [5,6,48]. HLA-DQ2 is a heterodimer encoded by different DQA1 (α chain) and DQB1 (β chain) genes. DQA1*0501 and DQB1*0201 make up DQ2.5, and DQA1*0201 and DQB1*0202 make up DQ2.2 [79,80]. DQ2.5+ antigen-presenting cells (APCs) have a higher bound peptide stability and longer gluten presentation than DQ2.2+ APCs. Such differences in stability are associated with the risk of developing CD. Individuals with the DQ2.5 haplotype thus have a higher risk of developing CD than those with the DQ2 [80]. The DQ2.5 heterodimer is the most important heterodimer in CD and is present in approximately 90% of CD patients [3]. The DQ2.5 heterodimer may be encoded in either cis or trans gene configurations, both of which are associated with CD [79,80,81]. The DQ2/DQ8 prevalence in the general population is 30–40%, and only about 3% of carriers develop CD [82]. The prevalence of DQ2/DQ8 is higher in CD patients than in the general population, with more than 99% of CD patients carrying DQ2 (≥90%) and/or DQ8 (5%) [3]. Although the presence of DQ2/DQ8 genotypes is essential for disease development, it is not sufficient alone, and other genes at non-HLA loci must also be involved in addition to environmental factors. DQ2/DQ8 account for 35% of genetic susceptibility in the development of CD, and non-HLA genes contribute about 65% to genetic susceptibility [83].

#### 3.1.2. The Incidence of HLA-DQ2 and/or DQ8 in the Far East

The frequency of DQ2 in Western European Caucasian populations has been estimated at 20% to 30%, with relatively high frequencies occurring in North and West Africa, the Middle East, and Central Asia [38]. There is geographic variation in the frequency of DQ2 carriers. The frequency of DQ2 decreases from west to east, becoming less frequent in Southeast Asia and Japan than in more western regions [38]. Reported DQ2 carrier frequencies are less than 5% in Japan, South Korea, the Philippines, and Indonesia; 5–20% in China, Mongolia, Singapore, Taiwan, Thailand, and Vietnam; and >20% in Pakistan, Iran, Israel, and Saudi Arabia [38]. DQ8 carriers are globally distributed and have no geographical characteristics. Focusing on Far Eastern countries, the frequency of DQ8 carriers is 8–10% in Japan, which is similar to Western countries [84,85]. However, the DQ2 carrier frequency in Japan is extremely low at 0.3–0.6% [84,85]. A report found that the DQ8 carrier frequency in South Korea was 5–20% [86]. The DQ8 prevalence was reported to be 8% in China [87] but 20–25% in northwestern regions with non-Chinese minorities [8].

#### 3.1.3. Differences in Genetic Factors among Japanese

Of the 27 cases of CD and seropositivity reported in Japan, the HLA locus was investigated in 13, of which 9 (approximately 70%) were DQ2/DQ8-negative (Table 1). Rarely, a patient diagnosed with CD encodes neither the DQ2 nor the DQ8 heterodimer. Three large-scale studies in Europe, the United States, and Italy found that the prevalence of her DQ2/DQ8 negativity in CD patients ranged from 0.16% to 2% [88,89,90]. There have been several studies concerning the determination of CD risk development associated with the presence of the HLA-DQ genotype. Almeida et al. found that the risks associated with her DQ2.5/DQ2.5, DQ2.5/DQ2.2, and DQ2.5/DQ8 genotypes in the Brazilian population were 1:7, 1:10, and 1:19, respectively. The risk associated with having neither DQ2.2 nor DQ2.5 nor DQ8 was 1:3014. Studies from Italy and Syria highlight CD2 and CD8 as the greatest risk for developing CD, although genotype-related risks are somewhat different [81,89,91]. In southern Italy, 4.2% of CD patients were DQ2/DQ8-negative. Among patients with CD in southern Italy, DQ2/DQ8 negativity (38%) was significantly more frequent than DQ2/DQ8 positivity (24%) among CD patients. In DQ2/DQ8-negative CD, DQ7 was one of the most abundant haplotypes [92]. The haplotype HLA-DQA1*03-DQB1*03:03 (HLA-DQ9.3), which is common in the Chinese population but less so in Caucasians, is a genetic factor associated with CD susceptibility in China [93]. In Japan, many cases of CD lacked both DQ2 and DQ8, and approximately 80% of DQ2/DQ8-negative cases have DQ6 (Table 1). Further research and the accumulation of more cases are needed to determine if there is a unique CD susceptibility gene in Japanese as in Chinese individuals.

### 3.2. Gluten Exposure

#### 3.2.1. Gluten Exposure in CD Patients

The pathogenesis of CD at any age requires exposure to gluten. The CD prevalence can still be related to the consumption of cereal that contains gluten (mainly wheat) in the population [27]. The quantity of gluten in an infant’s diet may affect the risk for the clinical expression of CD or at least the earlier timing of its onset. In a prospective observational multinational study of more than 6605 children with genetic predisposition for CD due to their HLA antigen genotype, the quantity of gluten exposure during the first 5 years of life was associated with development of celiac autoimmunity and confirmed CD. By 3 years old, the absolute risk for developing celiac autoimmunity and CD was 28% and 21%, respectively, among children who consumed the reference amount of gluten (mean intake 3.7 g/day), compared with 34% and 28% among those who consumed an additional 1 g/day of gluten [94]. Time to the initiation of gluten ingestion, breastfeeding duration, and the avoidance of cow’s milk were not related to the risk of developing CD [95]. Current recommendations are to start giving gluten-containing foods to infants between 4 and 12 months old, as recommended in a position paper from ESPGHAN [96,97]. In a multicenter trial conducted throughout Italy, 533 infants were randomly assigned to groups that were introduced to gluten at 6 or 12 months old. By 10 years old, 16.8% had developed CD, with no significant difference between groups in disease development apart from a slightly delayed risk of CD in the 12-month-old group [98]. In a multicenter randomized trial, 944 infants were randomly assigned to groups given low-dose daily gluten or placebo at 4 months old, followed by the full introduction of gluten at 6 months old in both groups. The prevalence of CD was 12.1% at 5 years old, with no significant difference between the age groups [99].

#### 3.2.2. Gluten Exposure in Japan

The dietary intake of wheat in Japan is only one-third of that in Western countries, although consumption of these products has increased in recent years [38]. The consumption of rice in Japan has halved over the past 50 years, and the westernization of the diet is increasing [100,101]. However, Japanese guidelines for infants recommend eating porridge instead of wheat as the main grain, and porridge is a popular weaning food [102]. Although the exact intake amount of wheat products during infancy is unknown, Japanese food allergy guidelines list “udon” as the standard wheat product for childhood food load testing, indicating that it is widely accepted by the Japanese [103]. A study measuring wheat gluten content in South Korea found that strong flour contained 51.2–86.9 g/kg, and soft flour contained 23.0–47.3 g/kg. The gluten content of noodles was the highest in “soba” (43.2–72.6 g/kg) and lowest in “udon” (6.5–30.3 g/kg) [104]. The gluten content of white bread is 24.0–40.2 g/kg [104]. Therefore, the gluten content of udon is considered to be equal to or lower than that of white bread. The wheat protein concentration of the udon noodles in the above report is unknown. In Japan, udon noodles often use flour with a wheat protein content of about 9%, and when boiled, they absorb a lot of water. Therefore, the amount of wheat gluten per gram would be expected to be less than the same amount of bread. In Sweden, Germany, and Finland, which are European countries with a high incidence of CD, the amount of gluten consumed by a 1-year-old child is ≥2 g per day [37]. There is no accurate information concerning gluten exposure during infancy in Japan, but rice is a staple food, so gluten exposure during infancy may not be high among Japanese children.

#### 3.2.3. Other Factors

Environmental risk factors for CD aside from gluten intake have also been anticipated. It is likely that some risk factors, such as gluten introduction, are induced in early childhood, whereas others (such as smoking) contribute to risk later in life. Perinatal factors, including being small for gestational age and elective (but not emergent) cesarean birth, have been associated with a modest increase in the risk of CD, although data concerning the effects of cesarean birth are mixed [105,106]. Infections with microbes, such as rotavirus [107], Campylobacter species [108], or reovirus [109], might trigger CD, and vaccination against rotavirus might be protective [110]. Aspects of the gastric environment, including a lack of Helicobacter pylori colonization, are associated with an increased risk of subsequent CD. Helicobacter pylori infection rates were lower in CD patients than non-CD patients, suggesting that Helicobacter pylori infection may be a protective factor [111]. Genetically predisposed patients may also be more likely to develop CD following SARS-CoV-2 infection, making COVID-19 a candidate culprit for a potential outbreak of CD in the near future. Notably, however, COVID-19 patients with CD have no increased risk of hospitalization, death, thrombosis, or need for intensive care unit care compared to COVID-19 patients without CD [112]. Antibiotics and proton pump inhibitors exposure during infancy increase the risk of developing CD [113]. A nonsmoking status [114] and socioeconomic factors, including more maternal education, have been associated with an increased risk of CD [115], although the mechanism underlying this association remains unclear. The current research was mainly focused on the role of microbiota in the development of CD in children. A few recent studies have shown the altered microbiota in these patients and explained the association of microbiota and development of the immune system in CD [116]. To our knowledge, however, there has been no evidence of any unique factor that makes Japanese people more likely to develop CD than others.

## 4. Presentation

### 4.1. Clinical Manifestations of CD

The clinical manifestations of CD are variable and involve multiple organ systems. Symptoms can be broadly classified as intestinal or extraintestinal, and the symptoms that appear vary according to age [117] (Table 3). In the past, CD usually presented in infants and young children with malabsorption and failure to thrive. Recently, however, CD has tended to present later, between 10 and 40 years old, with milder gastrointestinal or nongastrointestinal manifestations [118]. This change in the presentation may be due to the increased recognition of asymptomatic and mild cases due to advances in serological screening [30]. The distinctive difference between children and adults is the clinical manifestations at the time of the diagnosis (Table 3). Symptoms in infants are usually different from those in older children. Diarrhea, anorexia, abdominal distension, and abdominal pain are usually seen in younger children. If the diagnosis is delayed, failure to thrive, irritability, and severe malnutrition can be seen. Gastrointestinal symptoms, such as diarrhea, nausea, vomiting, abdominal pain, abdominal distension, weight loss, and constipation, may occur in older children, depending on the gluten intake [53,119]. Extraintestinal symptoms are frequent in adult CD cases and may appear associated with other digestive symptoms, such as asthenia, oral sores, osteoporosis, or skin lesions [117,120]. CD patients may present extraintestinal manifestations as well as neurological disorders, often with seropositivity for antineuronal and antiganglioside antibodies [121]. Patients with CD can show a delay in the diagnosis, with a median delay of one year for children and more than four years for adults [4,122].

### 4.2. Classification of CD

CD is subtyped into four different categories based on the presentation as per the Oslo classification: classical (typical), nonclassical (atypical), subclinical, and potential CD [26,83]. 

Classical CD includes three features. The first characteristic is the presence of signs of malabsorption, such as steatorrhea, weight loss, or other signs of nutrient or vitamin deficiency. The second is the presence of histologic changes on a biopsy, including villous atrophy. Third, mucosal lesions and symptoms caused by ingestion of gluten-containing foods improve within a few weeks to a few months. 

In nonclassical CD, symptoms of malabsorption are minimal or absent, and the usual symptom pattern includes isolated vomiting or constipation, recurrent abdominal pain, extraintestinal symptoms (Table 3). These are the predominant symptoms, especially in older children and adults [118]. 

Subclinical CD has no discernible symptoms of CD, instead being diagnosed based on a positive finding on a specific serologic test for CD and biopsy evidence of villous atrophy. These cases are usually detected by screening of high-risk groups. 

Potential CD refers to cases with positive results for specific antibodies of CD (e.g., tTG antibodies) without biopsy data consistent with CD. Such patients are at a high risk of developing CD.

### 4.3. Presentation of Japanese Patients with CD

Of the 27 cases of CD and seropositivity reported in Japan, 11 were serologically diagnosed with asymptomatic occult CD, 5 with classical CD, and 11 with nonclassical CD (Table 1). Surprisingly, there have been no pediatric cases of CD reported in Japan. Therefore, an atypical course that differs from typical symptoms should be noted. 

## 5. The CD Diagnosis

### 5.1. Diagnostic Method

The diagnosis of CD is based on a high index of clinical suspicion, serological markers, and small bowel biopsies. CD guidelines recommend serological screening for CD in patients with questionable symptoms or high-risk patients, provided they are on a gluten-containing diet. Screening of asymptomatic patients without risk factors is generally not recommended. Serological results and the degree of clinical suspicion determine whether to proceed with an endoscopic biopsy. A tentative diagnosis of CD is made in patients with a positive result of a specific antibody test and characteristic histologic changes in the intestinal mucosa on a gluten-containing diet. If a gluten-free diet normalizes antibodies and improves symptoms, the diagnosis can be confirmed. NASPGHAN suggests small bowel biopsies in all suspected cases, whereas the ESPGHAN has proposed diagnostic criteria that excluded a biopsy [5,28].

### 5.2. Serological Testing

One of the most common screening tests for the diagnosis of CD is the detection of IgA tTG while consuming a normal diet. The sensitivity and specificity of tTG is excellent (95% and 96%, respectively) for identifying CD cases, and it is currently recommended as a screening test by both the NASPGHAN and ESPGHAN. In children <2 years old, however, tTG is not very sensitive [124], and antideamidated gliadin antibody and anti-EMA are preferred. In cases with IgA deficiency, IgG-based tTG and deamidated antigliadin antibodies are the recommended markers, although the sensitivity is not high [5,6,48]. A gluten-free diet increases false negative results, so the test should ideally be performed during a period of consuming a gluten-containing diet [125]. Nice guidelines recommend that serologic testing for CD be followed on a gluten-containing diet for at least six weeks [126].

### 5.3. Biopsy Testing

Endoscopic small bowel biopsies are considered the gold standard for the diagnosis of CD [5,6,48]. Individuals with positive tTG-IgA or EMA-IgA findings should undergo an intestinal biopsy to confirm the diagnosis of CD. Multiple biopsies should be taken, including four from the distal duodenum and at least one from the duodenal bulb, as the disease may have a patchy distribution or initially be confined to the duodenal bulb. Biopsies are evaluated according to the modified Marsh–Oberhuber classification [127]. Results range from mild alterations characterized only by increased intraepithelial lymphocytes (Marsh type 1 lesion) to flat mucosa with total mucosal atrophy, complete loss of villi, enhanced epithelial apoptosis, and crypt hyperplasia (Marsh type 3 lesion) [3]. Marsh type 3 is classified as 3a, 3b, or 3c depending on the severity of villous atrophy [3]. Patients with abnormal findings (Marsh type ≥2) should undergo serology and treatment with a gluten-free diet to confirm CD.

### 5.4. Serological and Biopsy Testing in Japan

Reports of CD in Japan include cases diagnosed based on biopsy findings despite negative serological screening tests and improvement by a gluten-free diet (Table 1). This suggests that clinicians may not be adequately implementing the diagnostic criteria for CD and serological tests. Serological testing may be less frequently performed because it is not covered by health insurance, thus incurring costs, and is time-consuming because the test evaluations are outsourced overseas. The endoscopic appearance of CD is well recognized among Japanese gastroenterologists, and low-cost endoscopic testing is widely performed by well-trained endoscopists [100].

## 6. Standard Therapy

### 6.1. Gluten-Free Diet and Post-Treatment Follow-Up

The standard therapy for CD is the lifelong elimination of gluten from the diet. A strict gluten-free diet has been shown to improve symptoms, intestinal damage, and the quality of life [128]. Because gluten is contained in many foods in Western diets, a gluten-free diet requires significant lifestyle changes. Gluten is a group of proteins present in wheat (gliadin), rye (secalin), and barley (hordein) [129]. Therefore, CD patients should avoid all foods containing wheat, rye, and barley as well as any other foods containing gluten (Table 4). Although some nutritional deficiencies can be corrected by dietary management of CD, gluten-free diets are often limited because they are low in protein and fiber and high in fat and salt [130]. Gluten misfeeding and refractory CD due to a lack of attention are associated with the persistence of micronutrient and vitamin deficiencies (iron, vitamin D, vitamin B12, zinc, calcium, magnesium, selenium, vitamin C, and vitamin D) [131]. Gluten-free labeling may vary from country to country. Standards set by the International Codex Alimentarius Commission (Codex) and the Food and Drug Administration (FDA) allow manufacturers to label foods as “gluten-free” if the gluten content is less than 20 mg/kg [132]. The lower limit of gliadin intake that can cause mucosal damage is 10–100 mg/day [133]. It is unlikely that a gluten intake of less than 10 mg/d will cause significant histological abnormalities [134]. In children, a rapid improvement of clinical symptoms is seen within two to four weeks of diet alteration, and serological and histological responses are modest compared to clinical symptoms [135]. Histologic mucosal healing is confirmed in 95% of children within 2 years but in only 60% of adults, showing that healing takes a longer time for adults. [134,136]. The best marker under proper follow-up and management is the decline in the antibody levels. The presence of persistent positive antibodies usually indicates continuing intestinal damage with gluten exposure. Serological follow-up should be done within 6 and 12 months after the diagnosis and then once a year thereafter [137]. tTG-IgA is reported to be the best marker to monitor [138]. Patients who strictly adhere to a gluten-free diet have an average “time to return to normal tTG testing” of one year [139]. Complications usually occur in CD patients older than 50 years old and in patients who do not adhere to a strict gluten-free diet. Mortality in these patients is higher than in the general population [140], but late complications are rare at <1% [141]. Late complications of CD include refractory CD, intestinal lymphoma, small bowel adenocarcinoma, and ulcerative jejunitis [142]. Patients with persistent symptoms with histologic changes, such as villous atrophy on a small bowel biopsy after 6 to 12 months on a gluten-free diet, are said to have refractory CD. Such patients require strong medical management and treatment by gastroenterologists [6,48,126]. 

### 6.2. Problems with Gluten-Free Diets in Japan

A gluten-free diet is not easy to adhere to, even for Japanese patients, as the general population is consuming more wheat as time passes, with the diet becoming more Westernized [17]. There are a number of issues involved in implementing a gluten-free diet in Japan. The first is that not many restaurants provide gluten-free cuisine. Although the number of healthy adults who follow a gluten-free diet is increasing, there are still few restaurants that readily offer gluten-free meals. Second, soy sauce, a common ingredient in Japanese cooking, contains wheat. Soy sauce is a traditional Japanese fermented food and is used in a large number of Japanese dishes. There are five types of soy sauce (koikuchi, usukuchi, tamari, shiro, and saishikomi) that differ greatly in the combination and ratio of soybeans and wheat [145]. Tamari contains almost no wheat, while shirokuchi has more wheat than soybeans. However, when the amount of gluten in several commercial soy sauces was measured, no gluten was detected in the commercial soy sauces [146]. This indicates that gluten may have been broken down during the fermentation process, although the data are insufficient to conclude whether or not soy sauce contains any gluten peptides at all. Since soy sauce is used only as a seasoning in many Japanese dishes, the actual amount of soy sauce in the diet would be small, and the amount of gluten would be even smaller; the decision to consume soy sauce would therefore have to be based on the severity of the disease in the individual. Third, due to the small number of patients with CD, there are few physicians and nutritionists with extensive experience in nutritional guidance for celiac patients. Finally, there are differences in labeling standards for wheat products between foreign countries and Japan. The FDA and Codex standards for gluten-free diets specify a gluten content of 20 mg/dL [132]. Japan’s Food Labeling Law requires that processed foods in containers be labeled only for wheat in gluten-containing foods [147]. Therefore, if CD patients consume a processed food product sold in Japan that is not labeled for wheat, they may be consuming a food that still contains barley or rye. These problems may be resolved in the future as the Japanese population becomes more aware of gluten-free diets.

## Figures and Tables

**Figure 1 ijms-24-02075-f001:**
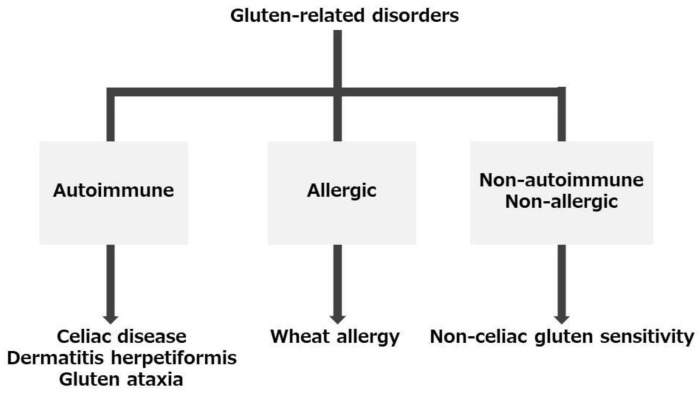
Immune reactions involved in gluten-related disorders. Gluten-related disorders are classified into three categories: autoimmune, allergic, and neither allergic nor autoimmune. Autoimmune disorders include celiac disease, dermatitis herpetiformis, and gluten ataxia. Allergic disorders are food allergy to wheat. Although the pathogenesis of nonceliac gluten sensitivity is not fully understood, it is considered to be nonautoimmune and nonallergic. The figure was created with reference to Sharma et al. and Taraghikhah et al. [1,2].

**Table 1 ijms-24-02075-t001:** The literature on cases of celiac disease and seropositivity in Japan.

Case No.	Years	Age (Years)	Sex	Symptoms	HLA Haplotypes	Serology (U/mL)				Biopsy (Marsh Class)	Associated Condition	Improvement on GFD	Classification of CD
						tTG -IgA	EMA -IgA	AGA -IgG	AGA -IgA				
1 ^a^	2022	47	M	Diarrhea Weight loss Malnutrition Edema	DQ2/DQ8	>100	NA	NA	NA	3c	Gastric cancer	+	Classical
2 ^b^	2021	68	M	Diarrhea Weight loss Malnutrition	DQ2/DQ8-negative	--	--	+	+	3	EATL UC	+	Classical
3 ^c^	2021	60’s	F	Diarrhea Stomach ache Nausea Malnutrition	DQ6	-	NA	NA	-	3b	-	+	Classical
4 ^d^	2021	37	F	Diarrhea Malaise Hypertransaminasemia	DQ4/DQ6	5	NA	10	38	3b	-	+	Nonclassical
5 ^e^	2020	38	M	Asymptomatic	NA	13.4	NA	NA	NA	NA	NA	NA	Potential
6 ^e^	2020	56	M	Asymptomatic	NA	11.3	NA	NA	NA	NA	NA	NA	Potential
7 ^e^	2020	90	M	Asymptomatic	NA	13.2	NA	NA	NA	NA	NA	NA	Potential
8 ^e^	2020	77	F	Asymptomatic	NA	12.8	NA	NA	NA	NA	NA	NA	Potential
9 ^f^	2019	45	M	Diarrhea Malaise Weight loss	DQ2	25	NA	NA	NA	3	EATL UC	+	Classical
10 ^g^	2018	66	M	Diarrhea	DQ4/6	12.9	-	NA	NA	3b	NA	NA	Nonclassical
11 ^g^	2018	72	M	Diarrhea	DQ6/7	29.8	-	NA	NA	3c	-	+	Nonclassical
12 ^g^	2018	58	M	Asymptomatic	DQ6/8	11.4	-	NA	NA	0	NA	NA	Potential
13 ^g^	2018	56	M	Asymptomatic	DQ7/9	20.6	-	NA	NA	0	NA	NA	Potential
14 ^g^	2018	52	M	Asymptomatic	DQ6/9	10.1	-	NA	NA	0	NA	NA	Potential
15 ^h^	2017	52	F	Diarrhea Seizure Ataxia Neuropathy Osteoporosis	NA	NA	NA	22.7	6.9	3	NA	+	Nonclassical
16 ^i^	2015	60’s	M	Diarrhea Abdominal pain Vomiting	DQ8	-	NA	NA	NA	3	-	+	Nonclassical
17 ^j^	2014	54	M	Diarrhea	NA	75.9	NA	NA	NA	3	Malignant lymphoma	+	Nonclassical
18 ^j^	2014	65	M	Diarrhea Anemia	NA	52.3	NA	NA	NA	3	Malignant lymphoma	+	Nonclassical
19 ^j^	2014	32	M	Diarrhea	NA	38.5	NA	NA	NA	3	Malignant lymphoma	NA	Potential
20 ^j^	2014	54	M	Asymptomatic	NA	12.5	NA	NA	NA	3	Malignant lymphoma	NA	Subclinical
21 ^j^	2014	67	M	Diarrhea	NA	10.5	NA	NA	NA	3	Malignant lymphoma	NA	Nonclassical
22 ^j^	2014	27	F	Asymptomatic	NA	21.7	NA	NA	NA	3	-	NA	Subclinical
23 ^j^	2014	70	M	Asymptomatic	NA	17.2	NA	NA	NA	3	Malignant lymphoma	NA	Subclinical
24 ^k^	2014	60’s	F	Diarrhea Weight loss Iron deficiency anemia	DQ6/6	-	NA	-	-	3	Malignant lymphoma	+	Nonclassical
25 ^m^	2008	68	M	Diarrhea Weight loss Edema Malnutrition	NA	38.2	NA	NA	NA	2 or 3	Malignant lymphoma	+	Classical
26 ^l^	2007	63	F	Diarrhea Vomiting Anemia	NA	NA	NA	NA	NA	3	EATL	NA	Nonclassical
27 ^n^	2006	65	M	Diarrhea Body edema Iron deficiency anemia	DQ6	52.3	NA	NA	NA	3	EATL	+	Nonclassical

^a^ Iwamoto et al. [11], ^b^ Miyagi [12], ^c^ Hayashida et al. [13], ^d^ Fujisawa et al. [14], ^e^ Fukunaga et al. [15], ^f^ Hiraga et al. [16], ^g^ Fukunaga et al. [17], ^h^ Sato et al. [18], ^i^ Baba et al. [19], ^j^ Nakazawa et al. [20], ^k^ Kishi et al. [21], ^m^ Makishima et al. [22], ^l^ Yasuoka et al. [23], ^n^ Makishima et al. [24].GFD: Gluten-free diet, EATL: enteropathy associated T-cell lymphoma, UC: ulcerative colitis, NA: not available.

**Table 2 ijms-24-02075-t002:** Global prevalence of celiac disease with a focus on Asia.

	Seroprevalence	Biopsy Prevalence
Global ^a^	1.4% (95% CI 1.1–1.7)	0.7% (95% CI 0.5–0.9)
By Continent		
Europe ^a^	1.3% (95% CI 1.1–1.5)	0.8% (95% CI 0.6–1.1)
North America ^a^	1.4% (95% CI 0.7–2.2)	0.5%
South America ^a^	1.3% (95% CI 0.5–2.5)	0.4% (95% CI 0.1–0.6)
Africa ^a^	1.1% (95% CI 0.4–2.2)	0.5% (95% CI 0.2–0.9)
Oceania ^a^	1.4% (95% CI 1.1–1.8)	0.8% (95% CI 0.2–1.7)
Asia ^a^	1.8% (95% CI 1–2.9)	0.6% (95% CI 0.4–0.8)
By Asian region		
Middle East ^b^	1.47% (95% CI 0.9–2.1%)	0.59% (95% CI 0.4–0.7%)
South Asia ^b^	1.25% (95% CI 0.6–2.5%)	0.87% (95% CI 0.4–1.5%)
East Asia ^b^	0.06% (95% CI 0.03–0.09%)	0.05% (95% CI 0.00–0.2%)

^a^ Singh et al. [7], ^b^ Ashtari et al. [41]. CI: confidence interval.

**Table 3 ijms-24-02075-t003:** Age related major clinical findings at the celiac disease diagnosis.

	Children < 2 Years Old	Children ≥ 2 Years Old	Adults
Gastrointestinal Symptoms	Diarrhea Vomiting Bloating	Loose stools Abdominal pain Dyspepsia	Dyspepsia Irritable bowel syndrome Constipation
Extraintestinal Symptoms	Malnutrition Failure to thrive Irritability Anemia	Iron deficiency Growth delay Headache Delayed puberty	Iron deficiency Osteoporosis Arthritis Hypertransaminasemia Recurrent mouth Ulcers Dental enamel effects Itchy blistering skin rash Neuropathy Ataxia Seizure

This table was created with reference to Vivas et al. and Anderson et al. [117,123].

**Table 4 ijms-24-02075-t004:** Fundamentals of a gluten-free diet.

Foods Containing Gluten		Foods with Possible Hidden Gluten
Wheat (Includes Kamut, Semolina, Spelled, Triticale) Rye Barley		Oats Fried foods (i.e., French fries) Beer Malt beverages Hard lemonade Coolers Breads Cereals Grains Cookies Cakes Croutons Gravies Imitation meats/seafood Self-basting poultry Salad dressings Sauces (i.e., Soy sauce) Canned soups Preservatives Food additives Products you handle Seasonings Medications Supplements
**Gluten free flour alternatives**		
Cereal grains	Amaranth, Buckwheat, Corn, Millet, Montina, Quinoa, Sorghum, Teff	
Legumes	Chickpeas, Kidney beans, Lentils, Peanuts, Navy beans, Pea beans, Soybeans	
Nuts	Almonds, Cashews, Chestnuts, Hazelnuts, Walnuts	
Seeds	Flax, Pumpkin, Sunflower	
Tubers	Arrowroot, Jicama, Potato, Tapioca, Taro	
Rice	Rice flour	

This table was created with reference to Harris et al. and Williams et al. [143,144].

## Data Availability

Literature supporting reported results can be found in the list of references.

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
