# Peer review of "Celiac Disease Genetics, Pathogenesis, and Standard Therapy for Japanese Patients"

_ijms, 2023, doi:10.3390/ijms24032075_

Round 1

Reviewer 1 Report

Since the prevalence of celiac disease (CD) in the Far East region of Asia, especially in Japan, is less reported compared to western countries, the authors outlined the epidemiology, diagnosis, and treatment of CD in this review manuscript. They summarize available the reported Japanese cases of CD with an overview of Japan.

The review is well presented and discussed, however, to highlight further the importance and need for increased diagnosis of CD, the authors should further stress some critical topics on extraintestinal manifestations as potential clinical presentation of CD, thus requiring to be screened for CD.

Regarding "High-risk groups", literature data on the atypical presentation of CD have reported that some conditions are significantly related to a higher risk of underlying CD. In particular, patients exhibiting "cryptogenic hypertransaminasemia" should be screened for CD as this condition may be the sole manifestation of silent celiac disease as previously reported (Anti tissue transglutaminase antibodies as predictors of silent coeliac disease in patients with hypertransaminasaemia of unknown origin. Dig Liver Dis. 2001;33(5):420-5.).

-Atopics subjects are also another higher-risk subgroup worth to be screened for CD diagnosis as previously demonstrated (Prevalence of silent coeliac disease in atopics. Dig Liver Dis. 2000;32(9):775-9.).

-Lastly, CD patients may present as extraintestinal manifestations also neurological disorders, often with seropositivity for antineuronal and anti-ganglioside antibodies as previously demonstrated (Sera of patients with celiac disease and neurologic disorders evoke a mitochondrial-dependent apoptosis in vitro. Gastroenterology. 2007 Jul;133(1):195-206; Anti-ganglioside antibodies in coeliac disease with neurological disorders. Dig Liver Dis. 2006 Mar;38(3):183-7.).

-Regarding "Diagnosis" the authors should also discuss current differences, if any, of CD diagnosis in ASIA and Japan, respst to other current international guidelines as well-summarized in a recent very comprehensive review (Anti-ganglioside antibodies in coeliac disease with neurological disorders. Dig Liver Dis. 2006 Mar;38(3):183-7.).

Reviewer 2 Report

Tamai and Ihara aimed to present an extensive review on Celiac Disease covering epidemiology, pathogenesis, clinical presentation, diagnosis and therapy also highlighting the specificities of the various aspects in Japan.

I have the following comments regarding the manuscript:

1.An English native should review the manus, as some of the paragraphs are not easily followed by the reader, because of some twisting of sentences and wording.

2. Page 4 – Epidemiology- line 5: “two peaks of onset: one immediately after weaning from gluten in the first two years of life”.  To my knowledge, celiac disease onset before age two is very rare among infants with gluten introduction not before 4 months of age. I would rephrase this and I would say – in the childhood, after two years of age.

3. Page 5- Epidemiology – Highrisk groups – line 6: “ In Japan, CD itself is rarely encountered even by general physicians, including those treating patients from high-risk groups,.. therefore …” . Please rephrase. I think, what the authors meant by the phrase was that general practitioners are not often enough taking into consideration Celiac Disease as a potential cause of the symptoms/ risk of the patients.

4. Page 6 – Down Syndrome – line 10: “However, there are no guidelines for CD screening in children with Down syndrome in other regsions or countries at present”. Please check EPSGHAN guidelines for the diagnosis of Celiac disease – 2012 -  at least must for sure contain recommendations on case finding strategy among Down syndrome children.

5. Page 7 – 2.3.9. IBD- line 1: “IBDs, including ulcerative colitis and Crohn’s disease, are associated with a high risk of developing CD.” I would suggest to change to “higher risk” even if the 0.93% reported prevalence is not higher than the one in general population which is about 1% and although in the cited study  the non-IBD patients showed a lower than general population.

6. Page 10 – Presentation – line 7: “Patients with CD can show a delay in the diagnosis, with a median delay of one year in children and a much longer delay of four years in adults” – The published diagnostic delay in adults is even longer than 4 years, as mentioned earlier in the manuscript by the authors. I would change to “more than 4 years”

7. Page 10 -  Table 3 – I would take out the “muscular atrophy” – is not characteristic for the vast majority of celiac disease children. When seen it is failure to thrive including muscular weakness, underdevelopment in general, including musculse, but I would not use the word “atrophy”.

8. Page 11 -  5.2 Serological testing – line 10: “A gluten intake of >3 g/day for 14 days induces histological and serological changes in the majority of adults with CD”. This is a controversial statement. There were  studies showing that 14 days are not enough. Please look at Lahdeaho paper and Sarna  paper.  Please refer to those studies also.
